# Regional Seismic Characterization of Shallow Subsoil of Northern Apulia (Southern Italy)

**Enrico Paolucci** [1,*]**, Giuseppe Cavuoto** [2]**, Giuseppe Cosentino** [3]**, Monia Coltella** [4]**, Maurizio Simionato** [4]**, Gian Paolo Cavinato** [4]**, Isabella Trulli** [5] **and Dario Albarello** [1,4]

1. Dipartimento di Scienze Fisiche, della Terra e dell'Ambiente, Università di Siena, 53100 Siena, Italy; dario.albarello@unisi.it
2. Istituto di Scienze del Patrimonio Culturale (ISPC), Consiglio Nazionale delle Ricerche (CNR), 80134 Naples, Italy; giuseppe.cavuoto@cnr.it
3. Istituto di Geoscienze e Georisorse (IGG), Consiglio Nazionale delle Ricerche (CNR), 56124 Pisa, Italy; giuseppe.cosentino@igg.cnr.it
4. Istituto di Geologia Ambientale e Geoingegneria (IGAG), Consiglio Nazionale delle Ricerche (CNR), Montelibretti, 00010 Rome, Italy; monia.coltella@igag.cnr.it (M.C.); maurizio.simionato@igag.cnr.it (M.S.); gianpaolo.cavinato@cnr.it (G.P.C.)
5. Autorità di Bacino Distrettuale dell'Appennino Meridionale, Valenzano, 70010 Bari, Italy; i.trulli@distrettoappenninomeridionale.it
* Correspondence: enrico.paolucci@unisi.it

**Abstract:** A first-order seismic characterization of Northern Apulia (Southern Italy) has been provided by considering geological information and outcomes of a low-cost geophysical survey. In particular, 403 single-station ambient vibration measurements (HVSR techniques) distributed within the main settlements of the area have been considered to extract representative patterns deduced by Principal Component Analysis. The joint interpretation of these pieces of information allows the identification of three main domains (Gargano Promontory, Bradanic Through and Southern Apennines Fold and Thrust Belt), each characterized by specific seismic resonance phenomena. In particular, the Bradanic Through is homogeneously characterized by low frequency (<1 Hz) resonance effects associated with relatively deep (>100 m) seismic impedance, which is contrasting corresponding to the buried Apulian carbonate platform and/or sandy horizons located within the Plio-Pleistocene deposits. In the remaining ones, relatively high frequency (>1 Hz) resonance phenomena are ubiquitous due to the presence of shallower impedance contrasts (<100 m), which do not always correspond to the top of the geological bedrock. These general indications may be useful for a preliminary regional characterization of seismic response in the study area, which can be helpful for an effective planning of more detailed studies targeted to engineering purposes.

**Keywords:** site effects; Northern Apulia; HVSR analysis; PCA

## 1. Introduction

It is well known that the seismostratigraphic configuration of the shallow subsoil (down to a depth of few hundreds of meters) may severely affect seismic ground motion due to constructive interference of seismic waves trapped within the surface and seismic impedance contrasts (e.g., [1]). This is why main efforts are devoted to promoting Seismic Microzonation studies as a basic tool to support city planning and land management policies aiming at reducing seismic risk at the municipal scale. For this purpose, a multiyear plan has been established in Italy to provide financial support to Seismic Microzonation studies in the areas characterized by higher seismic hazards [2]. Specific guidelines have been defined [3] to standardize these studies and to warrant their financial feasibility. To this purpose, three different levels of analysis are identified, each characterized by an increasing level of complexity and commitment. At the lowest level, the main efforts are devoted to fully exploiting the huge amount of data (drillings, geophysical surveys,

geotechnical reports, etc.) collected by local authorities and practitioners operating in the study area. Where this information is lacking or insufficient, geophysical surveys are planned by using cost-effective procedures based on ambient vibration monitoring, surface wave prospecting, etc. [4]. This information is considered to define a geological reference model (in the perspective of Seismic Microzonation) to be used as a basis for more quantitative evaluations provided by the higher levels of analysis (for details, see [2]). In this model, the soil configurations are classified in terms of engineering geological units by separating cover terrains and geological bedrock (for details, see [5]). It is worth noting the geological bedrock is defined in geological terms (age, tectonic relationships with overlying Quaternary sediments, etc.) with no reference to the stiffness of relevant rocks; thus, it may or may not correspond to the seismic (or engineering) bedrock considered by seismic rules (e.g., [6,7]).

A main target of the first level Microzonation, is the detection of seismic impedance contrasts relative to shear waves, which are the main source responsible for damage observed during earthquakes. However, in the lack of widespread direct geophysical observations (e.g., by borehole measurements) or where these contrasts are relatively deep (more than some tens of meters), their buried morphology must be inferred by geological and surface geophysical considerations (e.g., [8–10]).

An example of this situation is Northern Apulia (Southern Italy). This area, which extends for about 7000 km$^2$, has been affected in the past by large earthquakes with maximum effects larger or equal to IX MCS, which have been observed between 1361 AD and 1731 AD [11] (Figure 1). After that time, minor seismicity has only been observed in the area, and therefore, local authorities paid minor attention to the seismic characterization of this territory. Consequently, few geophysical data actually exist in the area, and this hampered the development of effective Microzonation studies for urbanized areas in this province. The aim of the present work is to fill this information gap by integrating available geological/geophysical information with new data obtained by an extensive geophysical survey based on a single station ambient vibration monitoring via Horizontal-to-Vertical Spectral Ratios (HVSR) approach (see, [12,13]). This large-scale survey was carried out on behalf of the local authorities (Autorità di Bacino Distrettuale dell'Appennino Meridionale) in cooperation with the Institute of Environmental Geology and Geoengineering of the National Research Council (CNR-IGAG) and the Department of Earth and Geoenvironmental Sciences of the University of Bari (Italy) [14].

The geological data available for the area in national, regional and local archives were collected and georeferenced (Figure 2A). The dataset includes 6 digitalized geological sheets at 1:50,000 scale from the CARG Project [15–20], 61 digitalized geolithological maps (scale 1:5000; for details, see [14]), 1401 well stratigraphy data of the Italian Law N. 464/84 database [21] (hereafter L464/84 wells), 107 well stratigraphy data of the ViDEPI Project for hydrocarbon exploration [22] and 1537 boreholes from continuous core drilling obtained from geotechnical reports. As concerns the geophysical information (Figure 2B), this dataset includes 50 down-hole and 5 cross-hole tests aimed at estimating the local shear wave velocity ($V_S$) profiles at a small set of sites; four of these cross-hole tests belong to the ITACA database [23]. Moreover, 60 seismic reflection lines of the ViDEPI Project [22] were also available.

In the following, the overall geological configuration of the study area is described at first. Then, the new data obtained by the extensive geophysical survey are illustrated, aiming at identifying the resonance phenomena responsible for the amplification of seismic ground motion. The collected data are then jointly interpreted with geological information to constrain the main seismostratigrahical features of the area.

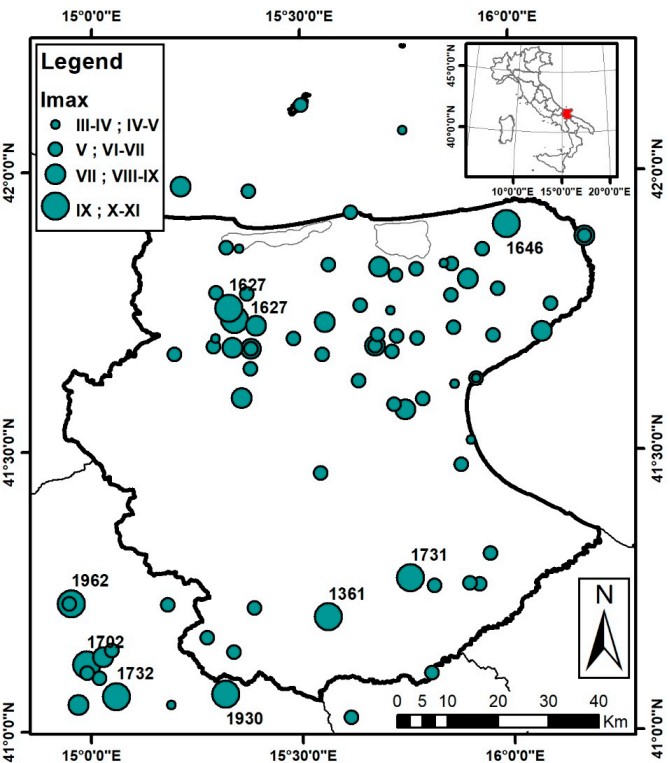

**Figure 1.** Dots represent epicenters of the main seismic events with maximum intensity (Imax) larger or equal to III-IV MCS that occurred in the study area and surroundings between 1000 and 2019 AD [11] (size is proportional to the relevant Imax value). The year of occurrence of the events characterized by Imax larger or equal to IX MCS is shown. The thick and thin black lines represent the border of the study area and the Italian regional administrative limits, respectively. The red area in the top right panel indicates the geographical position of the Northern Apulia area with respect to Italian territory.

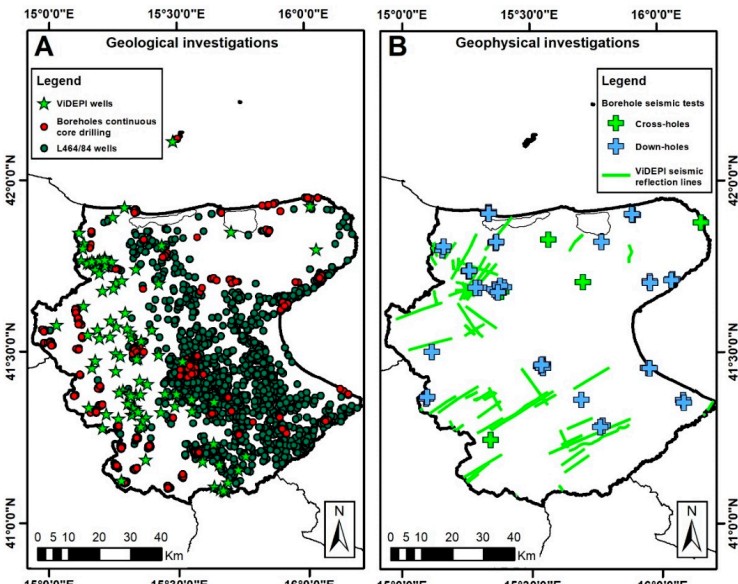

**Figure 2.** Location of the geological (**A**) and geophysical (**B**) investigations available in the study area. The thick and thin black lines represent the border of the study area and the Italian regional administrative limits, respectively.

## 2. Regional Geological Setting

The study area is located across three morphostructural domains as follows, from SW to NE, the classical chain-foredeep-foreland system: The Southern Apennines Fold and Thrust Belt, the Bradanic Trough and the Gargano Promontory of Apulia Foreland (Table 1, Figure 3). The Southern Apennines Fold and Thrust Belt (henceforth STFB) is a collisional belt represented by a complex tectonic assemblage of Mesozoic and Tertiary sedimentary units (Table 1) tectonically superimposed to each other (hereafter referred as "Allochthonous" Units), overthrusting the flexed foreland (the Apulian carbonate platform) outcropping in the Gargano Promontory (GP). In between, a foredeep basin (the Bradanic Trough, BT) was developed during the Plio-Pleistocene ages [24–26].

**Table 1.** Morphostructural domains and respective subdomains constituting the Northern Apulia. For each subdomain, the main features (geological-geomorphological description, age and maximum slope) are summarized together with a representative code also reported in Figure 3A.

| Morphostructural Domain | Subdomain | Code | Description | Age | Max Slope (Degrees °) |
|---|---|---|---|---|---|
| The Gargano Promontory of Apulia Foreland (GP) | Quaternary deposits | 1b | Alluvial/coastal plain deposits | Pleistocene-Holocene | 20° |
| | Mesozoic-Cenozoic Units of Apulian platform | 2 | Slope and basin carbonates | Low Cretaceous-Miocene | 43° |
| | Mesozoic Units of Apulian platform | 3 | Platform–Platform margin carbonate | Malm-Albian | 39° |
| Bradanic Trough (BT) | Quaternary deposits | 1a,1c | Alluvial/coastal plain deposits | Pleistocene-Holocene | 20° |
| | Bradanic Trough infill deposits (external foredeep depocenter) | 4 | Marine and continental sandy -fine deposits | Pliocene-Pleistocene | 19° |
| | Bradanic Trough infill deposits (internal foredeep area) | 5 | Marine and continental sandy -gravely deposits | Pliocene-Pleistocene | 26° |
| Southern Apennines Fold and Thrust Belt (SFTB) | Deposits of the Bradanic Trough onto Daunia Tectonic Unit | 6 | Marine and continental sandy -gravely deposits onto basinal and shelf margin facies | Pliocene-Pleistocene | 24° |
| | Daunia Tectonic Unit | 7 | Basinal and shelf margin facies | Oligocene-Late Messinian | 37° |
| | Fortore Tectonic Unit | 8 | Basinal facies | Late Cretaceous-Early Miocene | 33° |

Within these major domains, distinct subdomains can also be identified on the basis of peculiarities relative to structural setting, morphological features and lithostratigraphic configurations (Table 1). In the following, the main characteristics of these domains will be shortly outlined by focusing on the main relevant lithostratigraphic features.

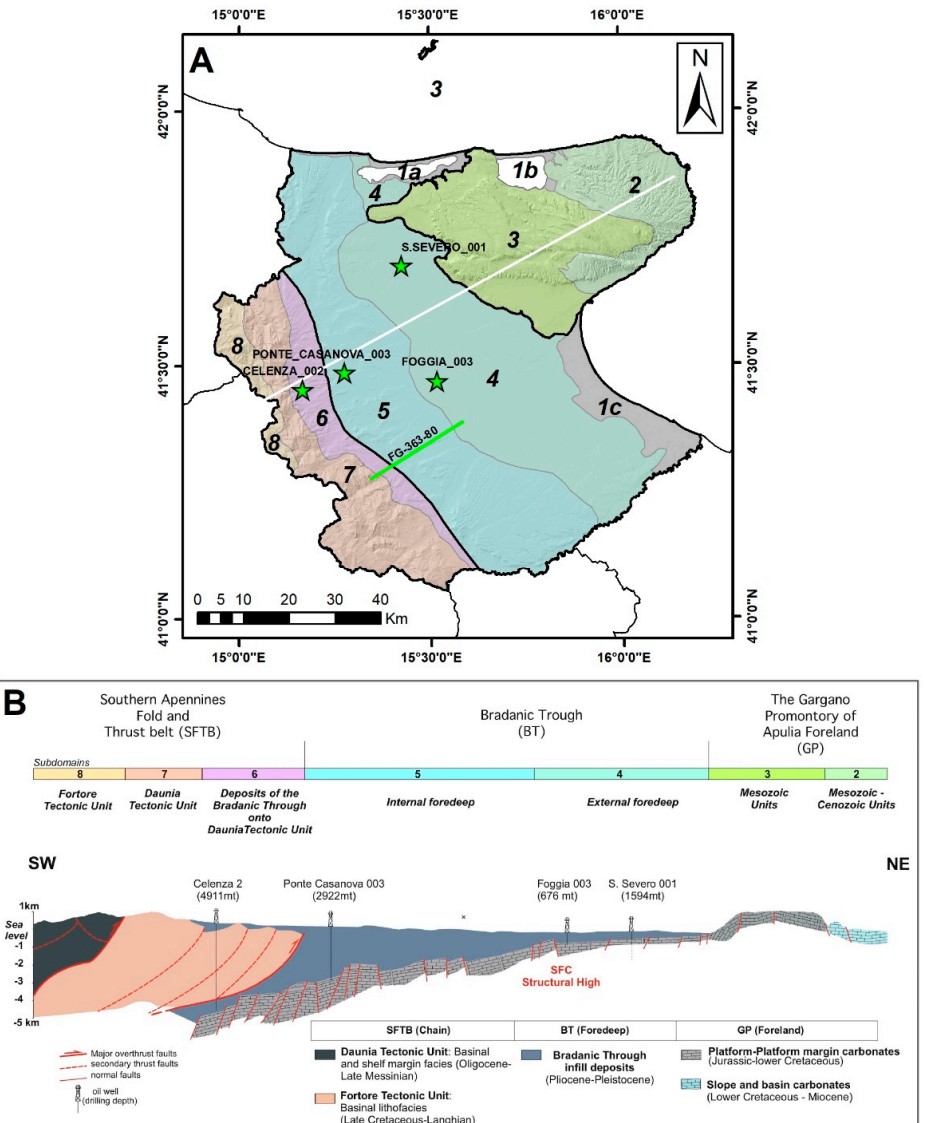

**Figure 3.** (**A**) Map of the morphostructural domains (delimited by thick black lines) and subdomains (colored zones delimited by gray lines) constituting the study area. In particular, subdomains 1b, 2 and 3 belong to the Gargano Promontory of Apulia Foreland; subdomains 1a, 1c, 4 and 5 belong to the Bradanic Trough; subdomains 6, 7 and 8 belong to the Southern Apennines Fold and Thrust belt. Green stars and green lines represent the ViDEPI Project investigations (Figure 2) used to constrain the schematic geological cross-section shown in panel B, whose trace is represented by the white line. Thin black lines represent the Italian regional administrative limits. (**B**) Schematic geological cross-section representing the overall configuration of the study area (modified from [16,27]).

## 2.1. Southern Apennines Fold and Thrust Belt (SFTB)

In this domain, the geological bedrock is generally composed of multilayered complexes with calcareous-marly-pelitic rocks of Cretaceous-Langhian age, followed by Miocene terrigenous arenitic-marly-pelitic alternations and Pliocene marine and alluvial units [28,29]. These rocky deposits can be covered by Quaternary eluvial-colluvial, alluvial pebbly-sandy-silty terraced and anthropogenic deposits.

All the bedrock units of this domain were strongly affected by intense tectonic deformation resulting in a highly weathered and fractured geological bedrock.

The western part of this bedrock is composed of Cretaceous-Miocene sedimentary successions, including limestones, marls, arenites, pelites and overconsolidated clays

covered by sedimentary deposits of the Quaternary period and can be divided into two regional tectonostratigraphic sectors.

In the inner-western sector (subdomain 8 in Figure 3), the geological bedrock is characterized by Late Cretaceous to Langhian, multilayered, calcareous-marly-pelitic and arenitic-marly-pelitic siliciclastic lithologies (Fortore Tectonic Unit [30]). In the outer-western sector (subdomain 7 in Figure 3), the bedrock is constituted by upper Cretaceous to early Miocene successions made by pelitic-calcareous and arenaceous deposits (Daunia Tectonic Unit [31,32]), passing to Lower-Middle Pliocene deposits. Finally, the eastern part of SFTB bedrock (subdomain 6 in Figure 3) is characterized by a buried thrust front deforming Miocene-Pleistocene deposits forming basins developed during the eastward migration of the tectonic front [33,34]. These basins are characterized by coarse continental and shallow silty-clay deposits.

### 2.2. The Bradanic Trough (BT)

It corresponds to the youngest morphostructural domain of the area [35,36] and represents a flexural depression developed at the front of the thrust belt during its eastward migration [37,38]. The overall thickness of the trough filling exceeds 3000 m [39,40]. The deepest part of the succession (Pliocene-Pleistocene in age) consists of a turbiditic complex [41,42], which lies on the Apulian carbonates. In particular, the Lower Pliocene part consists of a conglomeratic-sandy complex relative to the transgressive stage of infilling [43]. This thick marine succession underlying the Quaternary continental deposits represents the overconsolidated clayey-sandy bedrock. The Quaternary outcropping portion (i.e., the cover terrains) instead consists of a regressive succession of shallow-marine and/or continental-terraced deposits (Early–Late Pleistocene in age) that represent the upper part of the filling succession [44] lying on silty-clay deposits [43,45].

Data from the L464/84 wells and ViDEPI Project allowed the buried morphology of the carbonate bedrock to be reconstructed (Figure 4). Moving northeastward, it rises from a depth of approximately 3000 m to a depth lower than 200 m a few kilometers from the GP border (see also [46]). This inflected structure is bounded by extensional faults also affecting the overlying Lower Pliocene deposits that define a "horst-graben" structure (Figure 5). This geometry implies the presence of structural highs elongated in the NW–SE direction and rising from depths greater than 1000 to 250–350 m moving from the internal area to the foredeep central axis in the external area. Here, it is worth noting the NW–SE alignment of San Severo–Foggia–Cerignola (SFC in Figure 3B), where the carbonate bedrock is located in the depth range of 550–250 m [47].

As can be seen from the ViDEPI Project seismic reflection lines, the Pliocene-Quaternary infill, modulated by eustatic sea-level fluctuations, is characterized by several regional unconformity surfaces characterized by significant seismic impedance contrasts (Figure 5). They appear with a monoclinal trend slightly dipping eastward with an on-lap geometry on the Lower Pliocene deposits, which tend to taper eastward until they disappear in correspondence with the easternmost structural highs (SFC). The Middle Pliocene and Lower Pleistocene deposits can be ascribed to two distinct turbiditic cycles [24,41,45,48–50]. The lower turbiditic cycle is composed of non-channelized basinal sequences (high-efficiency turbidites; e.g., [51]) and includes the H4-H7 seismic horizons, while the upper cycle with channeled turbiditic sequences (low-efficiency turbidites; e.g., [51]) contains the H2 and H3 horizons that correspond to sandy horizons interposed in the turbiditic upper sequence. The H4 horizon corresponds to the boundary between the two depositional sequences, often marked by the presence of calcarenitic deposits. The shallowest seismic impedance horizon (H1) corresponds to the bottom of the Pleistocene depositional phase.

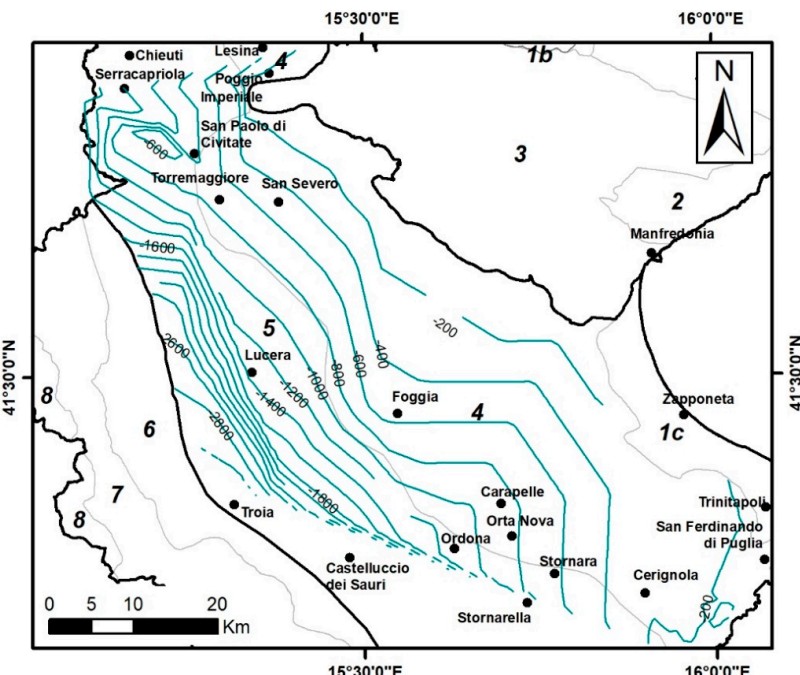

**Figure 4.** Map of the carbonate bedrock isobaths (blue lines) within the BT domain obtained interpolating the data from L464/84 wells and ViDEPI Project investigations. Thick black and grey lines represent the morphostructural domain and subdomain borders, respectively. Black points indicate the locations of the main urbanized areas of the Bradanic Trough.

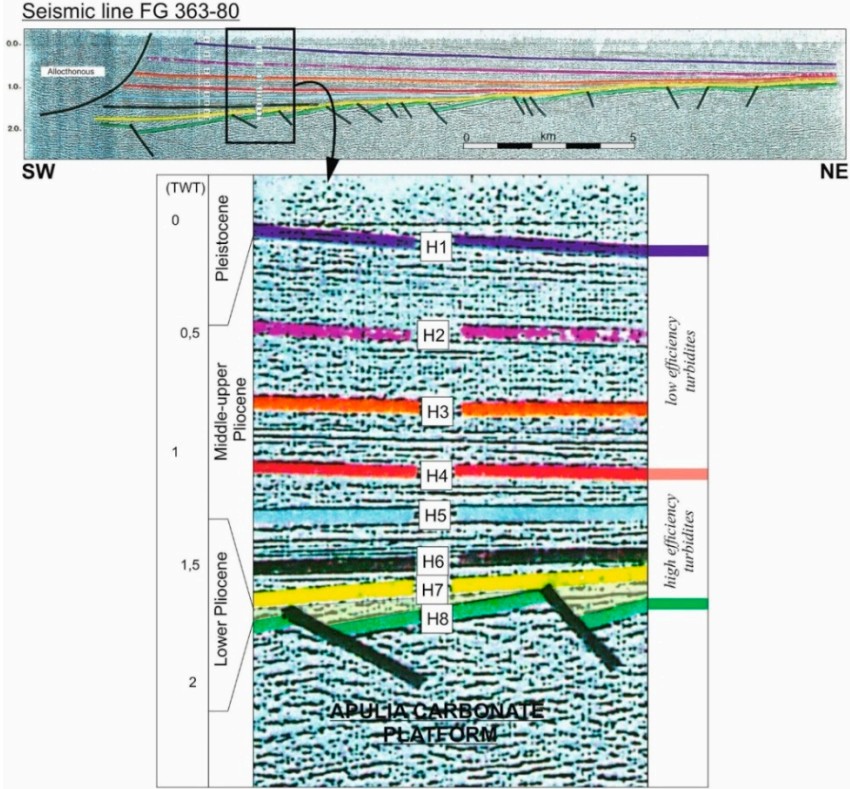

**Figure 5.** Line drawing and interpretation of FG 363–80 ViDEPI seismic reflection line (modified from [27]). The location is shown in Figure 3A. Colored lines represent the identified seismic horizons; thick black lines correspond to the recognized tectonic structure. TWT indicates the two-way travel time values.

Finally, an evident rise of the Apulian Platform can also be observed southeastward, where the carbonate bedrock reaches an 85 m depth near the SE limit of the BT.

Based on this information, in this morphostructural domain, three subdomains are recognizable (Table 1), characterized by distinct stratigraphical and structural signatures [52,53]:

- Internal foredeep area (subdomain 5 in Figure 3), characterized by a thick pile (2–3 km) of silty-clayey deposits covered by Quaternary very coarse-grained coastal and incised alluvial deposits that reach elevations of up to more than 600 m above sea level [54];
- external foredeep depocenter (subdomain 4 in Figure 3), where the top of the Apulia Platform rises to depths less than 1000 m and is covered by shallow-marine unit, ancient turbidite deposits, silty-clayey sediments and Quaternary continental and shallow-marine sandy-fine deposits;
- alluvial/coastal plains (subdomains 1a and 1c in Figure 3), which include transition areas. Subdomain 1a is a wide coastal area characterized by upper Pleistocene terraced alluvial deposits covered by Holocene marsh materials [55]. In this area, carbonate bedrock steeply deepens westward below the lagoon with horst and graben sequences, reaching depths greater than 200 m [56]. Subdomain 1c is a lacustrine relic plain, and it is characterized by Quaternary lagoon and alluvial deposits, together with limnic, marsh, beach and dune Holocene sediments [57–60]. These deposits reach a maximum thickness of 35 m and overlap with mud deposits.

### 2.3. The Gargano Promontory of Apulia Foreland (GP)

This domain corresponds to a structural high constituted of a thick succession of Mesozoic carbonates forming a gentle and asymmetric anticline dissected by an active and complex array and linkage of faults. In this tectonic context, "pull-apart basins" develop, linked to the strike-slip and extensional tectonics filled by late Pleistocene continental deposits shaped by processes of karstic origin [61–64]. This domain is morphologically characterized by a large central karst plateau with a succession of reliefs and depressions and karst landforms represented by dolines, structurally controlled poljes, and remnants of karst surfaces testify intense dissolution processes on the fracture network due to the intense tectonics.

In this domain, three distinct subdomains are distinguishable. In the first one (subdomain 2 in Figure 3), geological bedrock consists of carbonate units (Malm-Albian); moreover, Miocene-Pliocene calcarenites are present in the outer edges of the sector [65]. In the second one (subdomain 3 in Figure 3), the bedrock is characterized by slope and basinal carbonates of upper Cretaceous-Paleogene age [66–68]. Subdomain 3 bedrock is also characterized by Pleistocene carbonate conglomerate overlying the Mesozoic-Neogene carbonates [69].

Incisive karst processes are mainly set along the most important fault zone-related deformations [70] characterized by structural depressions filled by metric thicknesses of incoherent materials of medium-fine grain size, contained in pockets and cavities. These Quaternary deposits represent the cover terrains of the GP domain, together with ancient cemented groundwater debris, alluvial and lacustrine deposits, residual red soils and anthropogenic fill.

In the coastal plain located in the northern part of Gargano promontory (subdomain 1b in Figure 3) a lagoon bordered to the W and E by the limestone of Gargano platform is present. Similar to the subdomain 1a and 1c, this area is characterized by Quaternary coastal, alluvial and lagoon deposits [71].

### 2.4. Engineering Geological Characterization of the Cover Terrains

Based on information stored in the geological dataset, it was possible to estimate the minimum and maximum thicknesses of the engineering geological units (EGU) of the cover terrains (Quaternary deposits) in the three morphostructural domains and respective subdomains. As can be seen in Table 2, the thickness range of cover terrains in the SFTB

and GP domains is totally comparable (about 5–20 m) and significantly lower than the BT domain (18–41 m).

**Table 2.** Minimum thickness, maximum thickness and distribution in terms of percentage of the cover terrains for each of the three morphostructural domains and for the eight subdomains. Values in the rows named as "Total" were obtained by computing a weighted average, where the weight corresponds to the number of sites where the cover terrain characteristics were estimated within each subdomain.

| Morphostructural Domain | Subdomain | Minimum Thickness (m) | Maximum Thickness (m) | G% | S% | C% | O% |
|---|---|---|---|---|---|---|---|
| GP | 1b | 5 | 23 | 64 | - | 18 | 18 |
| | 2 | 6 | 19 | 72 | 14 | 14 | - |
| | 3 | 5 | 12 | 28 | 37 | 30 | 5 |
| | Total GP | 5 | 15 | 46 | 26 | 23 | 5 |
| BT | 1a | 30 | 40 | - | - | 29 | 71 |
| | 1c | 24 | 48 | 15 | 70 | 15 | - |
| | 4 | 22 | 50 | 22 | 46 | 32 | - |
| | 5 | 14 | 31 | 32 | 42 | 26 | - |
| | Total BT | 18 | 41 | 25 | 45 | 28 | 2 |
| SFTB | 6 | 3 | 12 | 67 | - | 33 | - |
| | 7 | 4 | 25 | 70 | 11 | 17 | 2 |
| | 8 | 3 | 12 | 61 | 8 | 31 | - |
| | Total SFTB | 4 | 23 | 70 | 11 | 18 | 1 |

G: gravels; S: sands; C: clays; O: organic soils; GP: Gargano Promontory of Apulia Foreland; BT: Bradanic Trough; SFTB: Southern Apennines Fold and Thrust belt. For the subdomain codes, see Table 1.

In order to grossly characterize the main properties of these terrains and geological bedrock EGUs, the engineering geological classification proposed by [72] has been adopted. In particular, four kinds of cover terrains are considered on the basis of the prevalent granulometry: gravels (G), sands (S), clays (C) and organic soils (O) (see Table 2). One can see (Table 2) that granular materials (consisting of gravels and sands) represent about 70–80% of the totality of cover terrains in almost all the domains and subdomains. In particular, the clear preponderance of the gravels is evident in GP and SFTB domains (in the latter, their percentage reaches as much as 70%), while the largest amounts of sandy materials in percentage terms are found within the BT domain. The remaining 20–30% of cover terrains consist of cohesive materials, represented by clays and organic soils: only in the 1a subdomain, these materials are predominant over the granular ones.

## 3. Single-Station Ambient Vibration Survey

### 3.1. The Geophysical Survey

An extensive geophysical survey was performed to integrate the available geological/geophysical information. In particular, single station ambient vibration measurements were carried out and considered to extract relevant Horizontal to Vertical Spectral Ratios (HVSR) to identify the seismic resonance phenomena induced by the presence of seismic impedance contrasts at depth (see, e.g., [73,74]).

Velocimetric acquisitions were carried out using the three-directional 24-bit digital tromograph Tromino™, produced by Moho SRL (Marghera, Venice, Italy; https://moho.world/, accessed on 30 August 2021) and the ambient vibrations were acquired for 20 min with a sampling frequency of 128 Hz. The HVSR curves were computed according to the procedure described by [13,75]. In particular, the spectra of the single components were computed by averaging 20-s-long non-overlapping windows; a detrend and a 5% cosine taper were applied to each window, and the spectra were smoothed by using a triangular

moving window with a frequency-dependent half-width (5% of central frequency in this case). Moreover, any window containing spurious signals was removed manually, and the quality of the resulting HVSR curve and the presence of unreliable experimental results were evaluated following the criteria described by [76].

As a whole, 403 ambient vibration single-station measurements were carried out. In particular, the measurements were deployed looking to homogeneously cover the main urbanized areas and, at the same time, to sample all the outcropping geological units to evaluate the possible different seismic responses within these areas. Overall, 10 measurements per municipality were performed on average.

### 3.2. Analysis of HVSR Data

By following [77,78], the collected HVSR curves were analyzed using the Principal Component Analysis (PCA). This automatic approach allows characteristic HVSR patterns within a set of measurements to be detected, as well as identifying and classifying sites where each of these patterns dominates the experimental outcome.

In particular, PCA was applied to the collected HVSR curves in the range of 0.1–10 Hz. The analysis shows that the first and most important Principal Component (PC) only accounts for 40% of the overall variance, and this suggests a rather heterogeneous subsoil configuration of the study area. It is worth noting that for each PC, two patterns are possible, here identified by the minus/plus sign. Following the procedure, 19 characteristic patterns were identified, and the 403 sites were classified, accounting for the locally dominant pattern (Table 3). Moreover, according to the adopted procedure, the sites where the HVSR curve indicates the absence of resonance phenomena ("no peak" in Table 3) are detected, and amplitude subclasses (Table 3) are defined. In particular, starting from the lowest to the highest, three levels of peak amplitude are evaluated: sites where weak seismic impedance contrasts are expected (subclass A, corresponding to HVSR amplitude around 2), sites where high (subclass B, HVSR amplitude 2–3) and very high (subclass C, HVSR amplitude larger than 3) impedance contrasts are expected. Despite their low amplitude, the peaks belonging to the subclass A are characterized by a physical plausibility (see, e.g., [13,76]), outlined by a localized lowering of the vertical amplitude spectral component with respect to the horizontal ones in correspondence of the peak frequency. This does not occur for the "no peak" class curves.

For each class and respective subclasses, the group average HVSR curves are computed: on the basis of these patterns, the peak frequency values of each class are identified. In particular, it was possible to distinguish the respective fundamental frequency value ($F_0$), defined as the frequency of the lowest frequency peak for each pattern, and (if any) the maxima at higher frequencies ($F_1$ and $F_2$, such that $F_0 < F_1 < F_2$). These values are summarized in Table 3.

Figures 6 and 7 show the experimental and the respective group-average curves (black and red curves, respectively) corresponding to the most populated classes by also considering the respective amplitude subclasses. As can be noted by Table 3, PC +1 and PC-1 classes dominate 101 and 105 sites, respectively, out of the 403 considered. Observing these two patterns, it is possible to see that the PC +1 class identifies one main peak at 5.5 Hz (Figure 6), while PC-1 is characterized by a multiple peak pattern: beyond the 0.3 Hz maximum ($F_0$), a secondary peak at 0.8 Hz ($F_1$) is present (Figure 7). Unlike the $F_0$ maximum, the amplitude value of the higher frequency peak remains about two in all the three subclasses, pointing out that a weak impedance contrast is possibly responsible for this effect. A similar situation can be observed for PC +2 and PC-2 classes: in the first case, only one peak at 2 Hz is detected (Figure 6), while the PC-2 class is characterized by a multiple peak pattern very similar to PC-1, where the only difference is given by the presence of a higher frequency peak located at 8 Hz ($F_2$) (Figure 7). As concerns the PC-4 class (Figure 6), group-average curves with broad peaks appear in the range of 0.65–1.2 Hz: by observing the experimental groupings, it is possible to note that the broad maximum actually results from a set of close sharp HVSR peaks included in this frequency range. It is

worth mentioning the case of the PC-6 class, whose pattern is characterized by a couple of peaks at 0.4 Hz and about 1.5 Hz (Figure 7). After a careful analysis of the individual component spectra, it is possible to realize that the higher frequency maximum is related to an industrial disturbance (e.g., [13,79,80]): the existence of this class testifies the capability of PCA in highlighting the HVSR curves affected by this kind of anthropic anomalies.

**Table 3.** Number of experimental HVSR curves (column Total) belonging to the classes identified by the procedure based on PCA. The term "no peak" denotes the class characterized by HVSR curves that indicate the absence of resonance phenomena. Columns A, B and C show the number of experimental curves belonging to the respective amplitude subclasses. In the last three columns, the peak frequency values identified by each group-average pattern are summarized. In particular, $F_0$ corresponds to the fundamental frequency value, i.e., the frequency of the lowest frequency peak; $F_1$ and $F_2$ represent the eventual higher frequency values, such that $F_0 < F_1 < F_2$.

| PC Class | A | B | C | Total | $F_0$ (Hz) | $F_1$ (Hz) | $F_2$ (Hz) |
|---|---|---|---|---|---|---|---|
| −1 | 24 | 65 | 16 | 105 | 0.3 | 0.8 | - |
| +1 | 26 | 41 | 34 | 101 | 5.5 | - | - |
| +2 | 28 | 25 | 9 | 62 | 2 | - | - |
| −2 | 5 | 14 | 17 | 36 | 0.3 | 0.8 | 8 |
| −4 | 5 | 16 | 6 | 27 | 0.65–1.2 | - | - |
| no peak | - | - | - | 15 | - | - | - |
| −3 | 6 | 5 | 3 | 14 | 4 | - | - |
| +3 | 5 | 8 | - | 13 | 1.5 | 8 | - |
| −6 | - | 6 | 5 | 11 | 0.4 | - | - |
| +10 | - | 2 | 2 | 4 | 10 | - | - |
| −10 | 1 | 2 | - | 3 | 8 | - | - |
| +5 | 1 | 1 | - | 2 | 0.6 | 3.5 | - |
| −5 | - | 2 | - | 2 | 0.35 | 2 | - |
| −9 | 1 | 1 | - | 2 | 2 | - | - |
| +7 | 1 | - | 1 | 2 | 1.5 | - | - |
| +6 | - | 1 | - | 1 | 0.25 | 0.9 | - |
| +4 | 1 | - | - | 1 | 2 | - | - |
| −14 | 1 | - | - | 1 | 3 | - | - |
| −16 | 1 | - | - | 1 | 2.5 | - | - |
| Total | 106 | 189 | 93 | 403 | - | - | - |

Finally, important information about the extent of the seismic impedance contrasts detected in the whole study area can be extracted from Table 3. In fact, observing the number of experimental curves belonging to the "no peak" class and A, B and C subclasses, it is possible to state that about 4% of the analyzed sites are not probably affected by seismic resonance phenomena, 26% identify weak impedance contrasts and about 70% are characterized by relevant impedance contrasts.

The map in Figure 8 shows the spatial distribution of the identified classes and subclasses in the study area. A good correlation is observed between the characteristic patterns and morphostructural domains. In fact, SFTB and GP domains present a marked prevalence of characteristic patterns with resonance frequencies above 1 Hz (mainly PC+1 and PC +2 patterns). As concerns the distribution of the amplitude subclasses, no particular spatial groupings are evident.

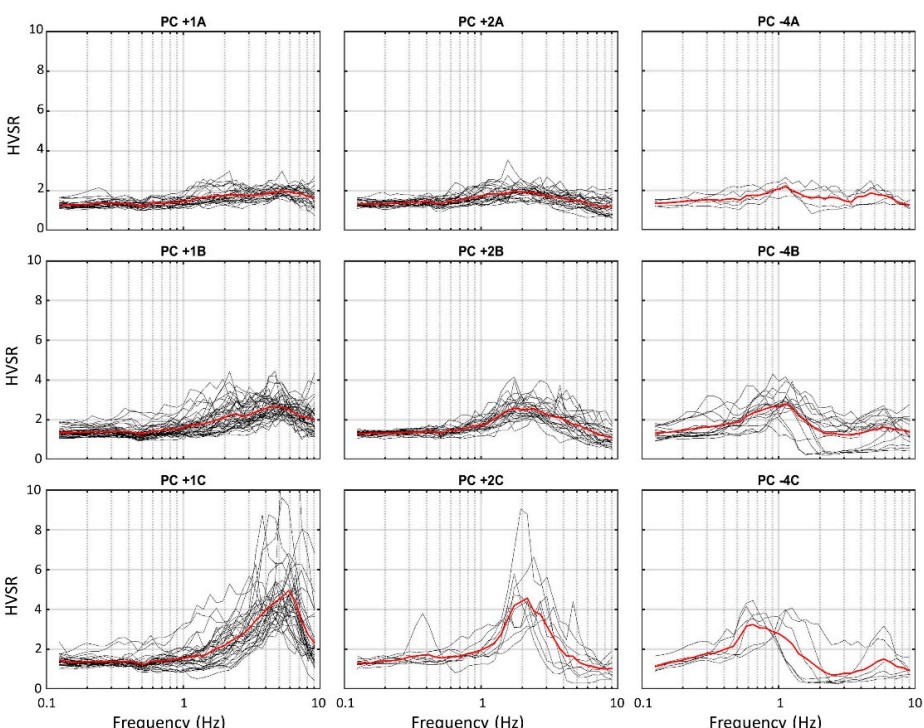

**Figure 6.** Experimental (black lines) and respective group-average (red lines) curves corresponding to the PC classes characterized by $F_0$ values in the frequency range of 0.65–5.5 Hz (PC+1, PC+2 and PC-4) by also considering the respective amplitude subclasses (A, B and C).

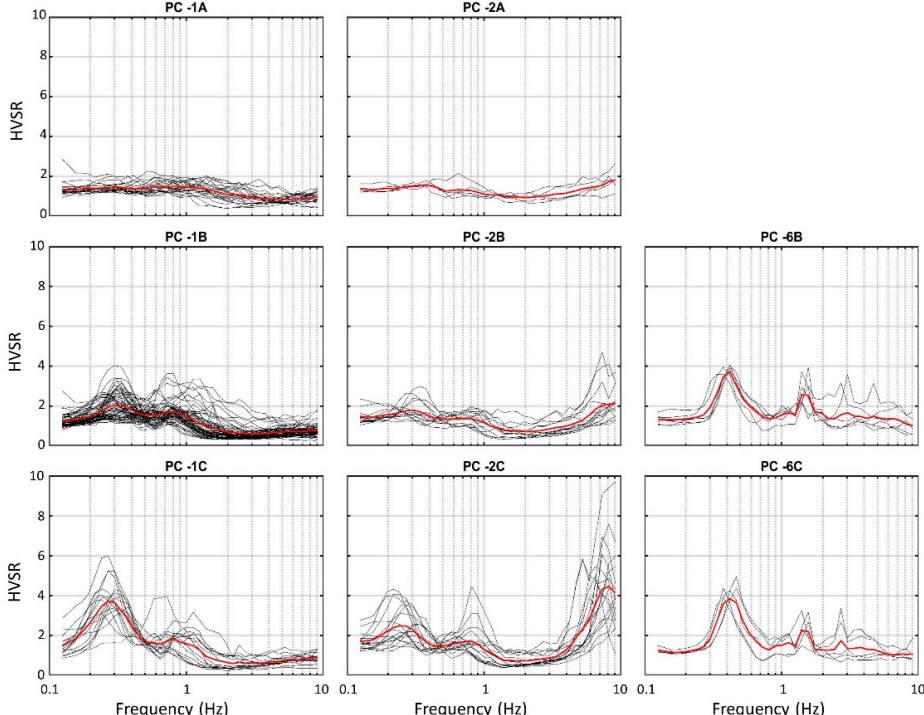

**Figure 7.** Experimental (black lines) and respective group-average (red lines) curves corresponding to the PC classes characterized by $F_0$ values in the frequency range of 0.3–0.4 Hz (PC-1, PC-2 and PC-6) by also considering the respective amplitude subclasses (A, B and C). The empty slot in the first row is due to the absence of HVSR curves belonging to subclass A within the PC-6 class.

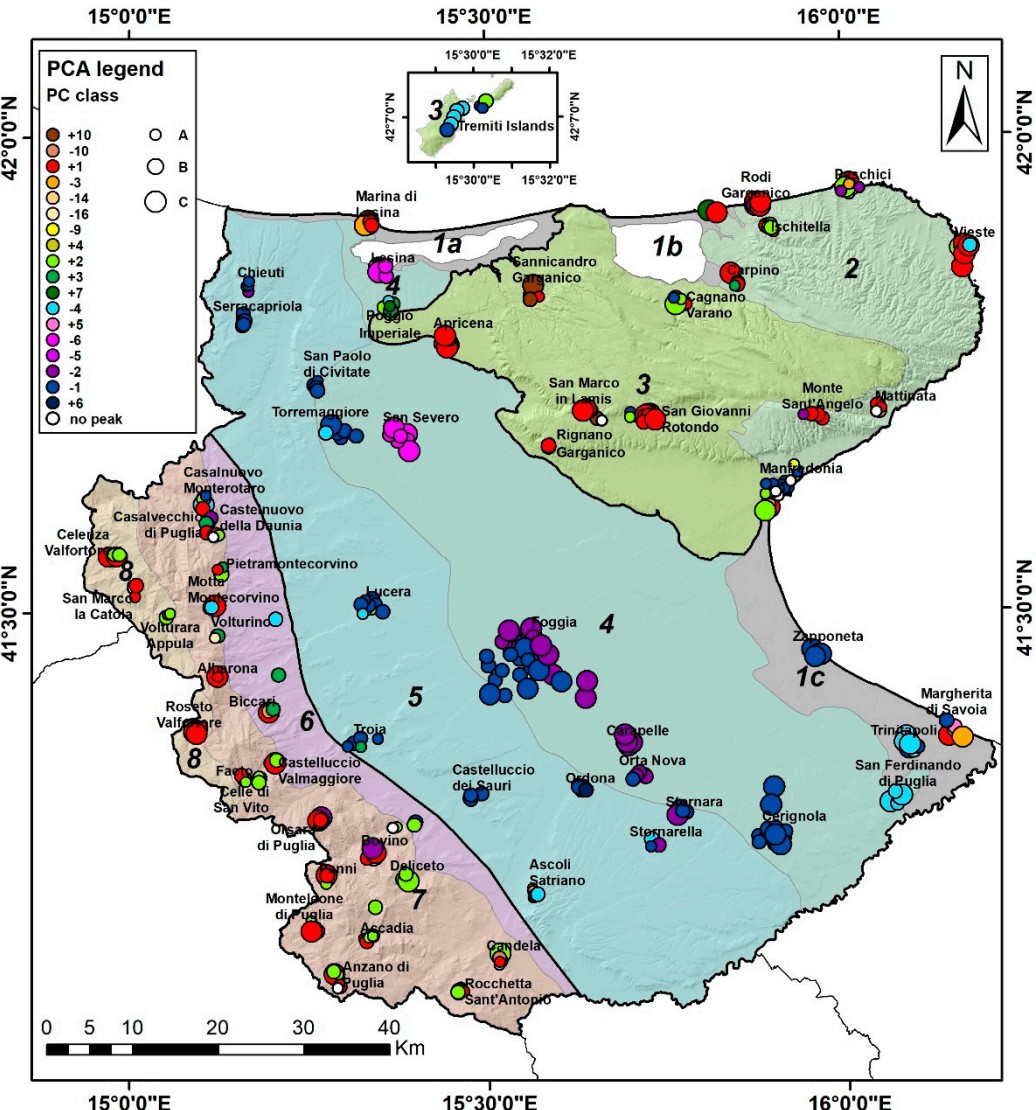

**Figure 8.** The dots in the map represent the locations of the single-station HVSR measurements, superimposed over the morphostructural domains (defined by the thick black lines) and respective subdomains (defined by colored polygons and respective representative codes) constituting the study area. The PC classes and their corresponding amplitude subclasses (A, B and C) are indicated by the color and size of the dots, respectively. In the PCA legend, the PC class names are sorted in descending order with respect to their fundamental resonance frequency ($F_0$).

On the contrary, the BT domain is characterized by the presence of characteristic patterns characterized by resonance frequencies below 1 Hz (mainly PC-1, PC-2, PC-4 and PC-6 patterns). In this last domain, some spatial clustering can be observed. In particular, sites characterized by the PC-2 pattern are mainly situated in the central part of the Bradanic Trough, within subdomain 4 (Foggia, Carapelle, Orta Nova, Stornara and Stornarella municipality); sites characterized by PC-4 pattern are mainly located in the SE part, between subdomain 1c and 4 (San Ferdinando di Puglia and Trinitapoli municipality); finally, sites where PC-6 pattern dominates are situated in the northern part of subdomain 4, within Lesina and San Severo municipalities: next to these two zones, the sources of the industrial disturbance at about 1.5 Hz characterizing this class are probably located. As concerns the amplitude subclasses, two different distributions can be observed in the PC-1 pattern within subdomains 4 and 5. In particular, the latter is characterized by a larger presence of relatively low impedance contrasts (subclass A), while in the first one, a greater

occurrence of B and C subclasses is found: this feature highlights the existence of a stronger impedance contrast within subdomain 4 related to the 0.3 Hz peak (Figure 7).

Finally, it is worth mentioning some minor class groupings showing differences with respect to the general pattern distributions described above. For example, within the GP domain, it is possible to note the PC-1A and PC-4 pattern groupings in Manfredonia and Tremiti Islands municipalities, respectively, and, within the BT domain, the presence of a group of sites where PC +1 and PC−3 patterns are located close to Margherita di Savoia and Marina di Lesina (in southeast and northern part of this domain, respectively).

Regarding the "no peak" measurements, this class is exclusively found in SFTB and GP domains, with a small grouping in Manfredonia municipality (subdomain 3).

## 4. Characterization of the Resonant Interfaces

### 4.1. Depth Estimate of the Resonant Interfaces

In order to associate a possible depth to the resonant interfaces responsible for the frequency peaks of the characteristic HVSR patterns, a simplified approach has been adopted (e.g., [76,81,82]). In this methodology, the velocity profile $V_S(h)$ above the resonant interface is tentatively assumed in the form of a power law:

$$V_S(h) = V_0(h+1)^x \tag{1}$$

where $V_S$ is the S-wave velocity at the depth $h$, and $V_0$ and $x$ are empirical parameters to be computed case-by-case. From Equation (1), one can see that the average velocity $\overline{V}_S(h)$ down to the depth $h$ has the form:

$$\overline{V}_S(h) \approx V_0(1-x)h^x \tag{2}$$

In this approximation, the depth $H$ of the resonant interface related to each resonance frequency $F$ identified by the PC classes can be estimated by an *F-H* relationship in the form [8]:

$$H \cong \left[ \frac{V_0\,(1-x)}{4F} + 1 \right]^{\frac{1}{1-x}} - 1 \tag{3}$$

To accomplish this procedure, the $V_S(h)$ profiles obtained by the 55 borehole seismic tests collected in the whole study area were considered. In particular, the $V_S(h)$ values related to the materials overlying the seismic bedrock (if geological bedrock characterized by a *Vs* value $\geq 800$ m/s has been identified during drilling) are considered. Following [82], the estimate of $V_0$ and $x$ values has been performed considering Equation (2) and produces the best fitting borehole $\overline{V}_S(h)$ profiles. These curves, computed in this case performing the harmonic mean up to any value of $h$, are generally smoother than the $V_S(h)$ profiles and this feature makes the fitting easier. Thus, a power law with parameters $V_0 = 245$ and $x = 0.16$ is obtained (Figure 9). Since the considered borehole seismic tests are mainly located within the BT domain (in particular, 41 out of 55), providing different sufficiently representative relationships for each morphostructural domain is not possible; thus, the estimated power law has been considered as representative for the whole study area. Even if, at first glance, it appears a strong approximation, this choice is justified by the composition of the cover terrains. In fact, as can be seen in Table 2, these terrains present a strong preponderance of granular materials (gravels and sands) compared to cohesive ones (clays and organic soils) in all three domains: from this characteristic, it is reasonably possible to hypothesize a similar lithostatic load effect affecting the materials overlying the resonant interfaces in the whole study area. Nevertheless, it was considered appropriate to estimate two further power-law patterns to bound the borehole $\overline{V}_S(h)$ profiles and capture their observed variability: in particular, lower and upper power laws are computed by subtracting and adding the regression standard error to the average power law, respectively. From this operation, we obtained lower and upper bounds marked by a curve with $V_0$ values

equal to 170 and 351, respectively; all three power laws are characterized by the same *x* value (Figure 9).

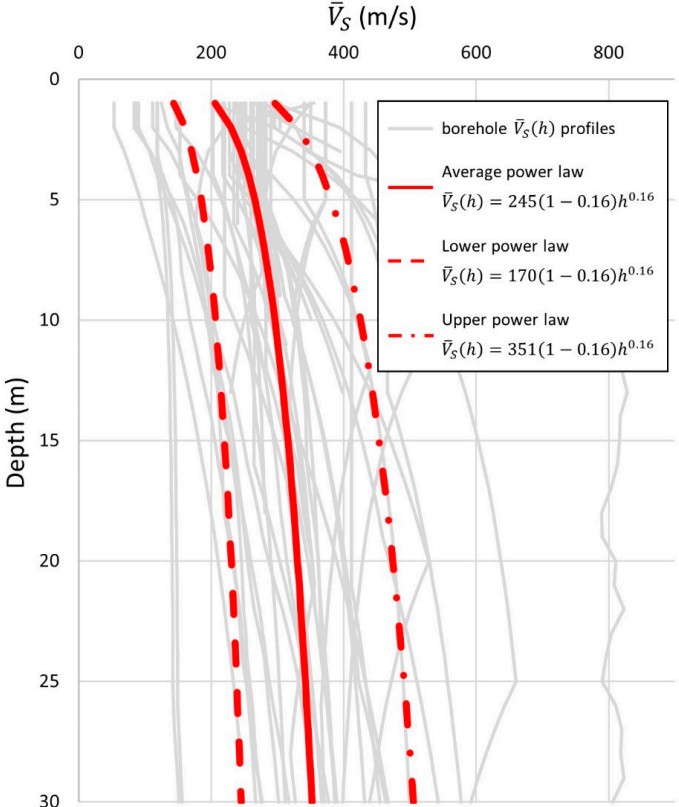

**Figure 9.** Borehole $\overline{V}_S(h)$ profiles (light grey curves) and the average (continuous red curve), lower (dashed red curve) and upper (dashed-dotted curve) power laws estimated to assess the depth variability of the identified resonant interfaces.

Finally, following Equation (3), for each resonance frequency value detected for each characteristic HVSR pattern, it was possible to compute a minimum, maximum and average depth of the resonant interfaces characterizing the whole study area (Table 4).

**Table 4.** Minimum (Min H), average (Avg H) and maximum (Max H) depth of the resonant interfaces responsible for the resonance frequencies (F) detected by the respective patterns of each PC class. Depth values are computed considering Equation (3) and, in particular, minimum, average and maximum depths are estimated using lower, average and upper power laws, respectively (Figure 9).

| F (Hz) | Min H (m) | Avg H (m) | Max H (m) | PC Classes |
|---|---|---|---|---|
| 10 | 5 | 8 | 11 | $F_0$ PC + 10 |
| 8 | 7 | 10 | 15 | $F_2$ PC − 2, $F_1$ PC + 3, $F_0$ PC − 10 |
| 5.5 | 10 | 15 | 23 | $F_0$ PC + 1 |
| 4 | 14 | 22 | 33 | $F_0$ PC − 3 |
| 3.5 | 17 | 25 | 38 | $F_1$ PC + 5 |
| 3 | 20 | 30 | 46 | $F_0$ PC − 14 |
| 2.5 | 25 | 37 | 57 | $F_0$ PC − 16 |
| 2 | 32 | 48 | 74 | $F_0$ PC + 2, $F_1$ PC − 5, $F_0$ PC + 4, $F_0$ PC − 9 |
| 1.5 | 44 | 68 | 104 | $F_0$ PC + 3, $F_0$ PC + 7 |
| 1.2 | 58 | 88 | 135 | $F_0$ PC − 4 |
| 1 | 71 | 109 | 167 | $F_0$ PC − 4 |
| 0.9 | 81 | 123 | 189 | $F_1$ PC + 6 |
| 0.8 | 93 | 142 | 217 | $F_1$ PC − 1, $F_1$ PC − 2 |

| F (Hz) | Min H (m) | Avg H (m) | Max H (m) | PC Classes |
|--------|-----------|-----------|-----------|------------|
| 0.65 | 118 | 181 | 277 | $F_0 \, PC - 4$ |
| 0.6 | 130 | 199 | 305 | $F_0 \, PC + 5$ |
| 0.4 | 210 | 321 | 492 | $F_0 \, PC - 6$ |
| 0.35 | 245 | 376 | 577 | $F_0 \, PC - 5$ |
| 0.3 | 294 | 451 | 692 | $F_0 \, PC - 1, F_0 \, PC - 2$ |
| 0.25 | 365 | 560 | 859 | $F_0 \, PC + 6$ |

### 4.2. Geological Interpretation

In the SFTB domain, resonant interfaces shallower than 20 m dominate. Observing Table 2, it is reasonable to assume that these interfaces, especially when they are associated with relevant impedance contrasts, correspond to the contact between the cover terrains (here mainly represented by Quaternary eluvial-colluvial and alluvial deposits) and the underlying bedrock units. Moreover, a significant contribution to seismic resonance phenomena in this domain is given by deeper resonance interfaces located in the depth range of 30–70 m: since the presence of cover terrains with these thicknesses is unlikely, these surfaces are probably related to lithological/structural variations or increase in the degree of compaction within the bedrock units. The same interpretation can be given to the seismic interfaces located at even greater depths (45–100 m and greater than 100 m) which rarely appear in subdomains 6 and 7. Finally, it is possible to note an association between outcropping geological bedrock and weak impedance contrasts or flat HVSR patterns.

A similar situation also occurs in the GP domain; resonant interfaces shallower than 20 m are predominant, and a significant presence of deeper interfaces at 30–70 m deep is found. As in the previous case, the first is associated with the contact between the cover terrains (here mainly represented by Quaternary residual soils, cemented ancient aquifer debris, silt-sandy alluvial/lacustrine deposits) and the limestone bedrock (Table 2), while the second can be referred to the transition between the fractured and weathered limestones and the more homogeneous and compact carbonate materials below. Very deep resonant interfaces (around and greater than 100 m) are detected in subdomain 3 in Isole Tremiti and Manfredonia municipalities, where intact bedrock outcrops: these surfaces, often associated with weak impedance contrasts, are related with undefined seismic interfaces within compact carbonate bedrock. The Manfredonia area is also characterized by a number of measurements that testify to the absence of resonance phenomena.

In the BT domain, an extremely different situation occurs. In particular, in almost the whole area, a very deep resonant interface is present, located in the depth range of 300–700 m, rising to 200–500 m in the San Severo and Lesina areas. Observing Figure 4, it is possible to note that the carbonate bedrock might be responsible for this low-frequency resonance effect in subdomain 4, especially along the SFC structural high. This interpretation can also be directly validated by the Foggia 003 and San Severo 001 ViDEPI oil wells (see the locations in Figure 3), where the carbonate bedrock is located at a depth of 585 and 385 m, respectively. This association is not valid within subdomain 5, where the Apulian Platform inflexes to reach depths greater than 1000 m (Figure 4). Considering the TWT (two-way travel time) values associated with the seismic horizons shown in Figure 5 and converting the $\overline{V}_S(h)$ profiles estimated by power laws in Figure 9 in average compressional wave velocity ($\overline{V}_P(h)$) profiles (assuming a value of 0.3 for the Poisson's ratio), it is possible to relate the resonant interface at 300–700 m depth to the H3 or H4 horizon. These two surfaces tend to taper eastward to disappear or overlap with the underlying seismic horizons (H7 and H8) in correspondence with the easternmost structural highs (Figure 5). The dual geological nature of this resonant interface within the BT domain can also be testified by the impedance contrast differences associated with the 0.3 Hz peak between subdomain 4 and 5: in fact, it is reasonable to assume that the stronger impedance

contrasts present in the first one are due to the contact between the carbonate bedrock and the overlying Plio-Pleistocene deposits.

Another deep, resonant interface exists in the BT domain area: this surface, always associated with weak impedance contrasts, is located in the depth range of 90–200 m, and it seems to disappear northward (in the proximity of the GP domain) and near the SE border. Exploiting the information shown in Figure 5 and following the above-described procedure for $\overline{V}_P(h)$ profile estimation, it is possible to associate this interface to the H1 seismic horizon, which roughly corresponds to the Upper Pliocene-Pleistocene transition.

Local significant rises of the Apulian Platform are related to the resonant surfaces located near the SE border and in the northern part of the BT domain. In particular, in the SE part, the carbonate bedrock rises from the depth range of 120–280 m (Trinitapoli municipality) to 60–140 m (San Ferdinando di Puglia Municipality), while it is situated at 30–100 m depth in the Poggio Imperiale municipality. These estimates are in keeping with those shown in Figure 4.

Finally, it is also worth noting the presence of resonant interfaces shallower than 20 and 30 m are located in Foggia-Stornarella with a NW–SE alignment (subdomain 4) and within the coastal areas (subdomains 1a and 1c), respectively. In the first case, the impedance contrasts are related to one or more gravelly horizons within the Quaternary continental cover, while in the second case to the contact between the Quaternary coastal deposits and the underlying geological bedrock.

## 5. Conclusions

The joint analysis of geological and geophysical data has been used for a first-order seismic characterization of Northern Apulia. The outcomes indicate that three main domains can be identified where seismic resonance phenomena are clearly differentiated. In the central domain (Bradanic Through), low frequency (<1 Hz) resonance phenomena uniformly dominate. These are associated with deep seismic impedance contrasts (depth >100 m) and the most significant ones corresponding to the top of the flexed carbonate bedrock in the easternmost part and, in the westernmost part, to a depositional transition within the Plio-Pleistocene infill of the through. As concerns the other two domains (Southern Apennines Fold and Thrust belt and Gargano Promontory), resonance generally occurs in the high-frequency range (>1 Hz), usually related to the bottom of cover terrains (i.e., the Quaternary deposits) and to the lithologic/structural changes within the geological bedrock.

This gross seismic classification of the study area should not be considered as a substitute for site-specific seismic response studies, which are mandatory for the anti-seismic design of structures. Due to the uneven distribution of observations and the use of a quite general (and very rough) procedure to infer the depth of resonant interfaces, our outcomes cannot be considered as fully reliable. Anyway, they provide general support for the correct planning of detailed seismic response studies. In particular, the identification of a generally low-frequency resonance in the whole Bradanic Through suggests that shallow investigation could be useless for a correct estimate of local seismic response since important seismic impedance contrasts only exist at depths larger than 100 m, well below the depth typically investigated (30 m) as suggested by current seismic rules (e.g., [6,7]).

Furthermore, the outcomes of the present study also suggest that the overall shape of HVSR curves joined with a gross engineering geological characterization of sedimentary covers, may be considered a proxy for a preliminary seismic classification of subsoil configurations in a target area. The study of similar proxies is an important research topic since they represent a key element of many seismic codes worldwide (see, e.g., [83–86]). In the case of the present approach, HVSR patterns representative of sets of measurements are considered in the frame of a PCA approach. This allows a more robust characterization of low strain seismic resonance phenomena with respect to single site assessment. On the other hand, this procedure does not allow site-specific characterizations. Moreover, no quantification of expected seismic effects is provided. This last aspect will require more

detailed and extensive studies (based on experimental and numerical analyses) that are well beyond the aims of the present work.

Finally, another important outcome is that the bottom of cover terrains (typically Quaternary deposits) in the three morphostructural domains does not represent the only significant seismic impedance contrast: consequently, a detailed mapping of this contact may not be enough for effective seismic characterization of the study area. For this purpose, the mapping of mechanical variations within the geological bedrock due to weathering/tectonic alteration (as in the case of Gargano Promontory) or lithological/structural transition (as in the case of Southern Apennines Fold and Thrust belt) plays an important role.

**Author Contributions:** Conceptualization, E.P., G.C. (Giuseppe Cavuoto) and D.A.; methodology, E.P., G.C. (Giuseppe Cavuoto) and D.A.; software, E.P. and G.C. (Giuseppe Cosentino); validation, M.C., M.S., G.P.C. and I.T.; investigation, E.P. and M.S.; writing—original draft preparation, E.P., G.C. (Giuseppe Cavuoto) and D.A.; writing—review and editing, E.P., G.C. (Giuseppe Cavuoto), D.A., G.C. (Giuseppe Cosentino), M.C., M.S., G.P.C. and I.T. All authors have read and agreed to the published version of the manuscript.

**Funding:** This research received no external funding.

**Institutional Review Board Statement:** Not applicable.

**Informed Consent Statement:** Not applicable.

**Data Availability Statement:** CARG Project geological sheets at 1:50,000 scale: https://www.isprambiente.gov.it/Media/carg/puglia.html (accessed on 30 August 2021); ViDEPI Project data: www.videpi.com (accessed on 30 August 2021); Italian Law N. 464/84 database: http://portalesgi.isprambiente.it/en (accessed on 30 August 2021).

**Acknowledgments:** Many thanks are due to the three anonymous referees whose suggestions helped us to significantly improve the original manuscript.

**Conflicts of Interest:** The authors declare no conflict of interest.

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
