# Peer review of "Regional Seismic Characterization of Shallow Subsoil of Northern Apulia (Southern Italy)"

_geosciences, doi:10.3390/geosciences11100416_

Round 1

Reviewer 1 Report

The manuscript "Large-scale seismic characterization of shallow subsoil of Northern Apulia (Southern Italy)" presents extensive research which contributes to seismic microzonation in this area. The ambient noise measurements are analysed using HVSR method and the data processing and interpretation are described well. Although the paper is weak regarding the novelty, I believe that it is interesting to the readers in the area of geological engineering and similar.

Below are comments that I hope will be useful for authors. Other specific comments and suggestions are given in the PDF file.

I suggest modification of the title in the sense that "Large-scale" is replaced with "Regional seismic characterization of shallow subsoil of Northern Apulia (Southern Italy)" or "Large-scale experiment for seismic characterization of shallow subsoil of Northern Apulia (Southern Italy)"

I recommend adding seismicity map of the area to Introduction.

In chapter 3.2 (Analysis of HVSR data) part in lines 335-355 is rather hard to follow. I suggest rewriting this part.

Reviewer 2 Report

The paper is dealing with a subject of great importance, the seismic characterization of shallow subsoils. In my opinion the manuscript fits the requirements for possible publication in Geosciences, after minor revision.

The authors collected an impressive amount of previous data (especially boreholes and surface and downhole seismic surveys) and they conducted numerous ambient noise measurements.

Below, I report my suggestions/comments for the manuscript.

1) The authors present a very detailed geological description of the regional geology with many details and geological terminologies which sometimes are not accessible to a non-geologist reader. I recommend to simplify and reduce the section 2: Regional geological setting. 

2) I suggest to import a better quality 2D cross-section in Fig.2B (I zoom in 200% to read, with difficulty again, the labels and legend of the figure). I recommend to import the orientation of the cross-section (as you made in Fig.4) and report the meaning of y-axis (which I suppose it corresponds to depth a.s.l.).

3) The authors should correct the depth of the bedrock isobaths at the bottom of Fig.3 close to Stornarella, Stornara, ... regions. I suppose the labels have been moved a little bit.

4) In table 2, I have not well understood the meaning of Total [GP, BT, SFTB].  I thought it was the average of the [Min, Max] thickness and the percentage of the geological materials, but it seems not stand, so please, explain it.

5) I am very skeptic how the results of the measurement points of the subclasses A & B, can really contribute to site response determination of the investigated area, since it is generally accepted among the scientific community that the HVSR ratios should have a minimum amplitude of 3, to considered as reliable. I don't know how sb can recognize any amplified peaks in the figures 5 & 6 for the subclasses A (mainly) & B, which appear with almost flat HVSR pattern! I imagine that the flat pattern will be the same at least for all the subclass A points; is that true? How different maybe the HVSR ratios of the "no peak" measurements with those of the A class measurements? I would like to see a "no peak" example.

6) I have not understood why the authors did not determine different V0 and x values for the 3 different domains [SFTB, BT, GP] and they used constant values assuming that the composition of their cover terrains is the same among them. The percentage between coarse grained and fine grained materials based on Table 2, are quite different among the three domains!

I will totally agree with the authors conclusion, that the results of the current work gives a gross idea for the seismic classification of the area and cannot substitute any site-specific seismic response study. It just gives a guide to further field surveys for site-specific seismic response characterization.

Reviewer 3 Report

The research was conducted to very good standard, and the research outcome is useful for earthquake engineering applications. Two specific comments: 

  1. It is well known within the earthquake engineering community that sediments down to only 30 m is not the best indicator of seismic site effects. However, it is simply hard to make changes to codes of practice.
  2. The authors should review the relevant works that have been conducted in other parts of the world, rather than focusing on works done within Italy only. Examples include: site classification scheme led developed by Kyriazis Pitilakis from Aristotle Uni Thessaloniki, Greece, advanced geophysical measurement techniques led developed by Michael Asten at Monash Uni, Australia, and site classification and soil resonance effect led investigated by Nelson Lam at Melbourne Uni, Australia. 
